# Probing the subtle differences between promethium and curium

Trenton B. Vogt [1,2,5], Megan E. Simms [3,5], Connor J. Parker [3], April J. Miller[3], Laetitia H. Delmau[3], Richard T. Mayes [4], Samantha K. Cary[3], Alyssa N. Gaiser [1,2], Cristian Celis-Barros [3] ✉ & Frankie D. White [3] ✉

Curium and promethium share similar chemical and physical properties thereby complicating their separation. Co-located processing at Oak Ridge National Laboratory results in curium contamination of the fission product stream containing promethium. To gain insight into the difficulty of this separation, the fundamental properties of these elements are experimentally and computationally probed in a 2,2':6',2''-terpyridine crystal system. Analysis of the isostructural compounds via single crystal X-ray diffraction and quantum theory of atoms in molecules reveals that bonding between promethium and curium is quite similar in this particular structure type. The small differences in the analysis of these two elements in this isostructural series sheds light on the difficulty required to separate the elements from each other. More so, this study develops the fundamental chemistries of two rare elements in the solid state and experimentally portrays the often-omitted position of promethium within the lanthanide series.

Since its discovery in 1945, promethium (Pm) remains the least studied element among the lanthanide series because of its scarcity and radiolytic nature. Previous investigations on milligram samples of Pm were performed decades ago using its most abundant isotope, [147]Pm. Many of these earlier studies examined the spectroscopic nature of simple [147]Pm compounds such as [147]Pm halide salts and oxides[1–5]. Compounds of [147]Pm typically are pale pink or purple in color, which is close to that of its neighboring lanthanide, neodymium (Nd)[6]. Unsurprisingly, like other lanthanide elements, [147]Pm possesses Laporte- forbidden f-f transitions in the visible region that give it a unique spectroscopic signature. The two prominent absorption peaks of [147]Pm appear at 548 nm and 568 nm and correspond to the $^5G_4 \leftarrow {}^5I_4$ and $^5G_2 \leftarrow {}^5I_4$ transitions, respectively. Like the spectroscopic studies, structural experiments involving Pm have been relegated to simple halide salts and oxides[7–9]. Interestingly, one study examined the phase change of metallic [147]Pm under pressure. The results of this study showed that the double hexagonal close-packed (dhcp) structure of metallic Pm switched to the face-centered cubic (fcc) structure type at 10 GPa, and

[147]Pm is the last metal in the lanthanide series that undergoes this phase transition with pressure[10]. Another isolated study examined the more complex organometallic structure, [147]PmCp$_3$ (Cp = cyclopentadiene). However, structural knowledge of this compound was measured by powder X-ray diffraction and only identified the unit cell parameters of the [147]Pm organometallic compound[11].

Today, the major application of [147]Pm is measuring the thickness of thin films by utilizing the β-decay property of [147]Pm[12]. However, [147]Pm has recently regained interest because of its potential use in lightweight betavoltaic batteries. With a half-life of 2.62 years, [147]Pm batteries could have a potential lifespan of ~five years, and [147]Pm has been previously demonstrated to work as a battery in pacemakers[13]. Unfortunately, the two primary pathways for producing [147]Pm are challenging. The first, and perhaps less efficient of the two methods, includes the irradiation of [146]Nd([146]Nd[n,γ][147]Nd($t_{1/2}$ = 10.98 d, β-). This method necessitates a challenging adjacent lanthanide purification of [147]Pm from large quantities of Nd coupled with long irradiation times to produce micrograms of [147]Pm[14,15]. The second, more feasible pathway,

[1]Department of Chemistry, Michigan State University, East Lansing, MI, USA. [2]Facility for Rare Isotope Beams, Michigan State University, East Lansing, MI, USA. [3]Radioisotope Science and Technology Division, Oak Ridge National Laboratory, Oak Ridge, TN, USA. [4]Nuclear Energy and Fuel Cycle Division, Oak Ridge National Laboratory, Oak Ridge, TN, USA. [5]These authors contributed equally: Trenton B. Vogt, Megan E. Simms. ✉e-mail: celisbarrosca@ornl.gov; whitefd@ornl.gov

comes from mining the waste stream of irradiated neptunium (Np) in the production of heat source [238]plutonium ([238]Pu) at Oak Ridge National Laboratory (ORNL), the only current producer of weighable quantities of [147]Pm[16]. This pathway, however, does not come without its challenges. Isolation of [147]Pm from the [238]Pu waste streams is confined to the shared hot cell processing facility of the Radiochemical Engineering and Development Center (REDC) at ORNL. Specifically, these hot cells are shared with the [252]californium ([252]Cf) production program which utilizes curium (Cm) for their target material. Because of the need to share processing equipment at ORNL, [244]Cm is present and coextracted with the [147]Pm product even after significant processing. The existence of [244]Cm in the [147]Pm material is problematic due to its neutron activity and fissile behavior. Currently, several solvent extractions of the [147]Pm material are required to remove the trace amounts of [244]Cm.

The coextraction of [244]Cm with [147]Pm in processing at ORNL suggests that the two elements have very similar chemistries and serve as the inspiration for this study. Both elements share the same dominant oxidation state (+3) and same predicted ionic radii (+3, VI-coordinate = 0.97 Å) thereby complicating the separation[17]. While there have been several studies examining the separation between Eu and Am[18–21], far less comparison studies, if any, exist between Cm and Pm. Furthermore, Pm chemistry is relatively unknown compared to the rest of the lanthanide series. Recently, an extended X-ray absorption fine structure (EXAFS) experiment has been performed on a Pm complex in nitric acid media to provide information on the bonding of [147]Pm in solution[22]. The results obtained were demonstrated with the ligand bipyrrolidine diglycolamide and measured [147]Pm−O bond distances of 2.476(16) Å, but the general chemistry of [147]Pm remains vastly unexplored. To further our expansion of Pm and Cm chemistry and obtain insight on the difficult [147]Pm/[244]Cm separation, we prepared two new isostructural compounds of [147]Pm and [248]Cm using the well characterized ligand, 2,2′:6′,2″-terpyridine (Terpy). The results provided solid-state bond distances of the Pm$^{3+}$ and Cm$^{3+}$ ions through single crystal X-ray diffraction (scXRD). The [147]Pm and Cm compounds are further characterized by spectroscopic and computational techniques to examine the fundamental differences between the two elements. Additionally, the work within also provides one of the few fundamental examples of how Pm resides within the lanthanide series, which is often omitted.

## Results and discussion

### Synthesis and crystallography

The prepared compounds of this study of the formula [M(Terpy)$_2$(NO$_3$)$_2$][M(Terpy)(NO$_3$)$_4$]·2MeCN (M = Ce − Er, Cm; MeCN = acetonitrile) were isostructural (excluding Ce) to those reported in literature[19]. The isostructural [147]Pm and Cm analogs along with several other unreported lanthanide analogs were prepared by mixing one

equivalent of the metal nitrate salt with three equivalents of Terpy in MeCN. Additionally, to minimize radioactive exposure, [248]Cm was used in place of [244]Cm. To obtain crystals suitable for scXRD, caution must be taken to avoid preparing a solution that is too concentrated, otherwise immediate precipitation occurs. Formation of crystals suitable for scXRD form between 2 and 48 h. In the case of [147]Pm, a prickly pear-colored solution (Fig. 1a) develops in which pale pinkish-purple crystals formed within two hours. However, these crystals quickly turned brown after a short period (~24 h) likely from radiolytic damage. All the metal compounds crystallized in the triclinic space group *P-1*. There are two distinct metal sites within the crystal structure. One site is a 10-coordinate metal site bound by two-tridentate Terpy ligands and two-bidentate nitrate anions to form a [M(Terpy)$_2$(NO$_3$)$_2$]$^{+1}$ (2:1 MTerpy) cationic species in the asymmetric unit. The second site consists of a 10-coordinate metal center bound by one-tridentate Terpy ligand and four nitrate anions. Three nitrates coordinate in a bidentate manner and one nitrate coordinates in a monodentate fashion to form a [M(Terpy)(NO$_3$)$_4$]$^{-1}$ (1:1 MTerpy) anionic species. The outer-sphere crystal environment is completed by two co-crystallized MeCN solvent molecules. Formulation of a nine-coordinate, tris-Terpy chelate system in which the metal site is completely saturated by Terpy ligands was not obtainable using the metal nitrate salts under these reaction conditions.

For PmTerpy, this structural system provides one of the first examples of scXRD Pm bond distances with an organic chelator. As expected, the bond distances for Pm lie between those observed for Nd and Sm. The average Pm−N bond distance in the cationic 2:1 PmTerpy species was observed to be 2.612(16) Å. This lines up nicely within the lanthanide series between NdTerpy [Nd−N$_{(Terpy)}$ = 2.621(17)] and SmTerpy [Sm−N$_{(Terpy)}$ = 2.599(17) Å]. Likewise, this expected trend was observed in the anionic 1:1 PmTerpy species within the crystal structure with an average Pm−N bond distance of 2.576(9) Å which lies in between the average Nd−N$_{(Terpy)}$ bond distance of 2.580(3) Å and Sm−N$_{(Terpy)}$ bond distance of 2.556(8) Å.

Given the generally accepted concept that 5f elements have a stronger preference for N-donor ligands than 4f elements, it was hypothesized that the Cm−N bond distances would be shorter than Pm although the two elements have the same predicted ionic radius. Albeit not statistically significant given the bond errors, this hypothesis appears supported in the MTerpy system in which the average Cm$_{(Terpy)}$−N bonds in the 2:1 cationic and 1:1 anionic metal site were 2.601(16) Å and 2.551(3) Å, respectively.

### Spectroscopy

The solid-state UV-Vis absorption spectrum of PmTerpy was measured alongside NdTerpy, SmTerpy, and CmTerpy (Fig. 2). The absorption spectrum of NdTerpy was as expected with the major absorption

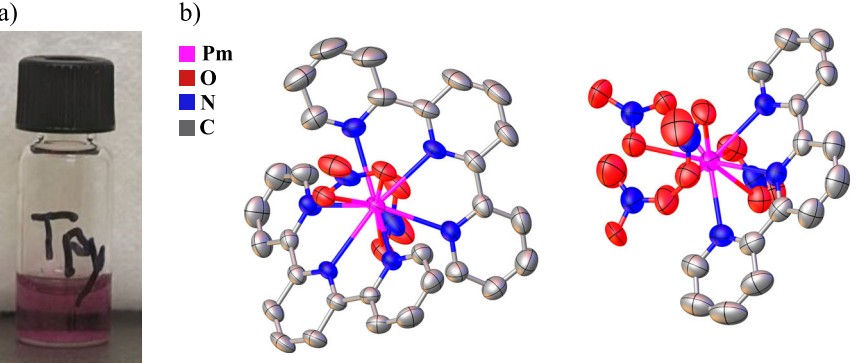

**Fig. 1 | The preparation of Pm single crystals. a** The reaction with [147]Pm and Terpy forms a prickly pear colored solution before crystallization. **b** The crystal structure of [[147]Pm(Terpy)$_2$(NO$_3$)$_2$][[147]Pm(Terpy)(NO$_3$)$_4$]·2MeCN (**PmTerpy**) shown at the 50% thermal ellipsoid probability. Hydrogen atoms and outer-sphere MeCN solvent molecules are omitted for clarity. Pink = Pm, red = O, blue = N, and gray = C in the crystal structure.

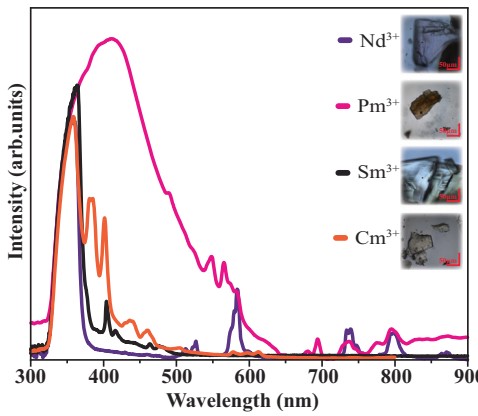

**Fig. 2 | The absorption spectra of MTerpy compounds (M = Nd, Pm, Sm, Cm).** The broadening of the peak at higher energy in **PmTerpy** is likely caused by the radiolytic damage observed in the crystal picture. Purple = Nd, pink = Pm, black = Sm, and orange = Cm. Red scale bars represent 50 μm.

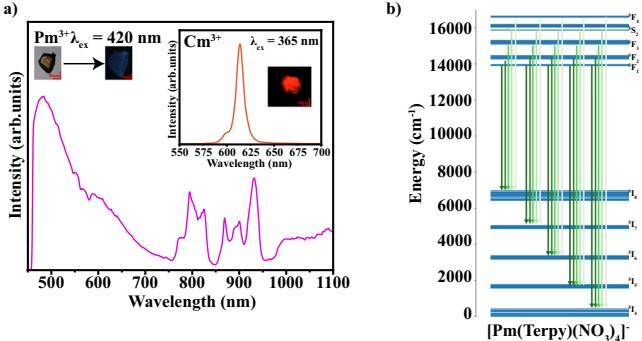

**Fig. 3 | The solid-state spectroscopic characterization of PmTerpy and CmTerpy. a** The emission spectrum of PmTerpy and CmTerpy (inlet). Pm exhibits a vibrant, complex emission spectrum. **b** The calculated LFDFT energy levels associated with the emission spectrum of the anionic 1:1 species in PmTerpy. The theoretical energy diagram was obtained from a TPSS-D3/STO-TZP LFDFT calculation. Red scale bars represent 50 μm.

bands occurring at 527 nm ($^4G_{7/2}$), 579 nm ($^4G_{5/2}$), 740 nm ($^4F_{7/2}$), and 795 ($^4F_{5/2}$) nm. Likewise, the major absorption peak at 404 nm ($^4P_{3/2}$) is observed in SmTerpy, although this peak occurs next to an intra-ligand transition most likely due to a $\pi \rightarrow \pi^*$ transition in the Terpy ligand. Like SmTerpy, CmTerpy has higher energy absorption peaks at 381 nm, 385 nm, and 401 nm which matches with absorption spectra of other solid-state Cm compounds reported in literature[23–25]. Additionally, the emission spectrum of CmTerpy was measured because $Cm^{3+}$ is known to have a characteristic emission peak from Laporte-forbidden f-f transitions. This peak is located at 614 nm (inset of Fig. 3) and matches closely with values reported for this singular peak in literature[23–25].

Although the solid-state absorption spectrum of the $Pm^{3+}$ ion was as expected, there are aspects worth discussing. At times, the color of lanthanide compounds can be attributed to their ligand environment, but their color often comes from their major absorption bands. For example, Nd compounds are often portrayed to be light purple in color because of the intense absorption bands between 508 nm and 575 nm. The most intense bands of $Pm^{3+}$ occur in a similar wavelength range to neodymium and would thus be expected to share a similar color to that of $Nd^{3+}$ or $Er^{3+}$. In the PmTerpy absorption spectra, the dominant peaks are located at 548 nm and 568 nm. These peaks lie between the major peaks observed for NdTerpy. However, while the color of the solution and crystals were initially prickly pear, the color of the PmTerpy crystals quickly turned brown (Fig. 2), likely from

radiation damage in the organic-containing compound. The expected Terpy-based transition at 360 nm is significantly broadened and red-shifted nm which could also be attributed to the radioactive nature of Pm. This effect has been observed in the solid-state absorbance of compounds containing berkelium (Bk), which is also a high specific activity β-emitter with a short half-life ($^{249}$Bk $t_{1/2} = 330$ d)[26].

Though some reports mention Pm luminescence peaks in the visible region[7], we could not confirm these peaks due to the broadband feature exhibited at higher energies and poor peak resolution. Other works propose Pm does not have peaks in the visible region[27]. Unfortunately, this area of Pm luminescence remains ambiguous and necessitates further study. Therefore, the focus is on the near-infrared (NIR) emission spectrum of Pm. The PmTerpy compound has a rich NIR emission spectrum with intense peaks at 797 nm, 828 nm, 872 nm, 903 nm, and 934 nm. All these peaks except the 872 nm peak agree with those computationally predicted using the ligand-field density functional theory (LFDFT) as shown in Fig. 3. This is likely due to emission from a higher-lying emissive state as reported elsewhere for $^{147}Pm^{3+}$ doped into a LaCl₃ chloride matrix[28]. Our calculations show that the multiplet structure of a $^{147}$Pm chloride complex and our Terpy system should not be significantly different and similar emission should be expected (Supplementary Fig. 1).

## Bonding

The structures of the 2:1 and 1:1 complexes were obtained computationally from the crystal structures (see Methods Computational details) with errors <2% in the M-N bond lengths (Supplementary Table 1). These structures were used to interrogate and compare the nature of the PmTerpy bonds to those of the Nd, Sm, and Cm analogs for the 2:1 and 1:1 complexes. The quantum theory of atoms in molecules (QTAIM) formalism has proven sufficient to this aim, where the electron density, energy densities, and delocalization indices have been the most popular metrics to shed light on the nature of the metal-ligand interaction. Table 1 summarizes the M-N$_{Terpy}$ average values for the selected QTAIM metrics. The accumulation of electron density is similar in all M-N bonds with Cm-N being slightly increased compared to the lanthanides. A similar trend is observed for the delocalization index. The negative values of the total energy density suggest that there is a non-negligible covalent interaction between the metal and Terpy ligands for all complexes. Though the overall metrics suggest a similar bonding pattern for all complexes, slight differences indicate an increased degree of covalency in Cm. However, it is noteworthy that the Nd-N bonds in the 2:1 show a more negative H(r) value ($-80.5$ kJ mol$^{-1}$ Å$^{-3}$) compared to that of Cm ($-76.8$ kJ mol$^{-1}$ Å$^{-3}$).

If we contrast these results with energy decomposition analyses, a similar behavior is observed (Table 2). For example, Cm shows a slight increase in orbital (2–5 kcal/mol) and total interaction energies (3–4 kcal/mol) compared to the rest of the lanthanide complexes.

**Table 1 | Averaged QTAIM metrics for the M-N bonds in the 1:1 and 2:1 complexes**

| | Bond | ρ(r) | G(r) | V(r) | H(r) | δ(r) |
|---|---|---|---|---|---|---|
| MTerpy 1:1 | Nd-N | 0.283 | 634.5 | -704.2 | -69.7 | 0.251 |
| | Pm-N | 0.285 | 658.1 | -720.5 | -62.4 | 0.253 |
| | Sm-N | 0.280 | 641.8 | -700.5 | -58.7 | 0.247 |
| | Cm-N | 0.320 | 771.3 | -859.2 | -87.9 | 0.274 |
| MTerpy 2:1 | Nd-N | 0.296 | 669.2 | -749.7 | -80.5 | 0.261 |
| | Pm-N | 0.281 | 642.9 | -702.3 | -59.4 | 0.245 |
| | Sm-N | 0.281 | 640.8 | -702.3 | -61.5 | 0.256 |
| | Cm-N | 0.309 | 738.8 | -815.6 | -76.8 | 0.265 |

Electron density, ρ(r), is given in e$^-$ Å$^{-3}$; kinetic (G), potential (V), and total energy (H) densities in kJ mol$^{-1}$ Å$^{-3}$. All the metrics on the table except the delocalization indices, δ(r), were calculated at the bond critical point.

Though the electrostatic component remains the most significant in magnitude, it is canceled out by the high Pauli repulsion frequently seen with neutral ligand fragments. This highlights the role of orbital interactions in these complexes.

The PmTerpy compound presented in this work provides crystallographic scXRD bond distances of Pm with an organic ligand. The reported Pm−N$_{avg}$ bond distances of 2.612(16) Å (2:1 PmTerpy) and 2.576(9) Å (1:1 PmTerpy) lie between Nd and Sm as expected. Likewise, The Pm−O$_{avg}$ bond distances to the nitrate anions of 2.55(3) Å (2:1 PmTerpy) and 2.54(6) Å also fall between NdTerpy and SmTerpy. Though expected, these results are a rare example of how Pm is experimentally positioned in the lanthanide series. While the Cm−N$_{avg}$ bond distances are slightly shorter than PmTerpy, the Cm·O$_{avg}$ bond distances of CmTerpy are similar for both the 2:1 and 1:1 species at 2.55(4) Å and 2.55(7) Å, respectively. This further hints at the general concept of 5f elements' preference to softer N-donors over the 4f elements.

Further examination of both Pm and Cm in this isostructural series reveals the true difficulty in separating the two elements. When conditions, ligands, or extractants are identical, like in processing, the two elements possess very similar chemistries. Many structural and chemical factors that make separations between elements feasible are absent in this situation. The structural differences between Pm and Cm in the crystal system discussed above are miniscule. The two elements share a similar bonding behavior, even at higher coordination numbers. Additionally, both elements' oxidation and reduction potentials to either the +2 or +4 state are not feasible in either ambient or aqueous processing conditions. While it was expected for Cm to have somewhat shorter bonds to the nitrogen-containing Terpy ligand, the differences are not significant. In this crystal system, Cm bond lengths align closely to Sm (Fig. 4). Therefore, purification of Cm from the Pm product is essentially in the realms of an adjacent lanthanide separation, which is extremely tedious. Oftentimes, computations may reveal significant differences

in bonding as observed in separation studies between Eu and Am[18–21]. However, only minute differences in bonding are shown by the computational calculations within this Terpy system. The lack of substantial difference in bonding (less than 3 kcal/mol between Pm and Cm orbital interaction, respectively), and similar bond lengths to Pm and Sm, could possibly be attributed to the ligand system itself and the reluctance of Cm$^{3+}$ (5f$^7$) to engage its 5f shell in enhanced covalent interaction.

The major differences between PmTerpy and CmTerpy arise in their spectroscopy. Of the actinides, Cm could be considered the most lanthanide-like because its half-filled 5f orbital makes obtaining other oxidation states challenging. This relegates Cm to almost solely existing in the +3-oxidation state in compounds other than oxides. Furthermore, Cm possesses a sharp luminescence spectrum in the visible range like europium (Eu), terbium (Tb), Sm, and dysprosium (Dy). Conversely, the major luminescence peaks of Pm are in the NIR region. The luminescence peaks in the visible region of Cm appear at higher energies, and these observed luminescence features could potentially be used in aiding the separation from each other by spectroscopically monitoring their elution from resins in a column purification as recently demonstrated for Cm[29].

In summary, the complicated separation of $^{147}$Pm from actinides and other fission products is critical for the long-term success of $^{147}$Pm production at ORNL. During the extraction of $^{147}$Pm, $^{244}$Cm is coextracted and necessitates multiple solvent extractions to remove trace amounts of the $^{244}$Cm neutron source from the $^{147}$Pm product. To shed light on this purification and further investigate the 4f/5f relationship, we prepared single crystals of both $^{147}$Pm and Cm in an isostructural lanthanide series with Terpy. Our results demonstrate the similarity between Pm and Cm while also signalizing subtle differences between the two elements. In the analyzed crystal structures, though Pm and Cm have the same predicted VI-coordinate ionic radius, Cm bond distances are not significantly shorter, which correlates with the reluctance of Cm(III) to deviate from the stable 5f$^7$ configuration. Additionally, the Cm bond distances align more so with Sm, which could make the separation of Pm and Cm like an adjacent lanthanide separation. These results were further supported by QTAIM and energy decomposition calculations, and the differences are not significant enough to emphatically enhance separations compared to other An/Ln N-donor systems[18–21]. The most observable difference between Pm and Cm in the Terpy structures arises from their luminescence, which could potentially be used as indicators on a chromatographic separation. Additionally, this work is one of the few current studies that fundamentally and experimentally positions Pm

**Table 2 | Energy decomposition analysis (kcal/mol) of the 1:1 MTerpy complexes**

|  | Nd | Pm | Sm | Cm |
|---|---|---|---|---|
| Pauli | 106.3 | 109.1 | 105.6 | 124.3 |
| Electrostatic | −105.2 | −108.7 | −105.6 | −121.4 |
| Orbital | −38.9 | −40.9 | −38.7 | −43.3 |
| Total | −37.8 | −40.5 | −38.7 | −40.4 |

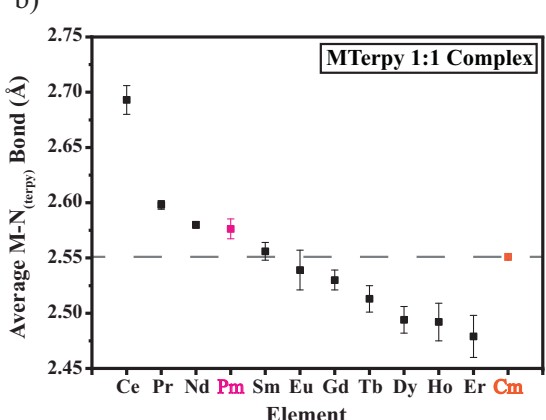

**Fig. 4 | The average M-N$_{(Terpy)}$ bond lengths of the isostructural complexes examined in this study.** In both the (**a**) 2:1 MTerpy and (**b**) 1:1 MTerpy crystal structure sites, slightly shorter Cm-N bond lengths (orange) are observed in comparison to Pm-N bond lengths (pink). The standard deviations in the averaged bond distance errors are also displayed.

within the lanthanide series and demonstrates the necessity for further study in the chemistry of these two rare elements.

## Methods

### Synthesis of crystal structures

Caution! $^{147}$Pm and $^{248}$Cm are radioactive in nature. $^{147}$Pm ($t_{1/2}$ = 2.62 y) has a high specific activity (928.4 Ci/g), and experiments with both elements were performed under the supervision of radiological control technicians (RCTs) in laboratories equipped with high-efficiency particulate air (HEPA) filters. $^{147}$Pm (>99% purity with less than 1% $^{147}$Sm and other impurities) and $^{248}$Cm (99.4% purity) were received at REDC through the National Isotope Development Center (NIDC) as the nitrate salts. All reagents and solvents were used as received from the manufacturer.

Stock solutions of $^{147}$Pm(NO$_3$)$_3$ and $^{248}$Cm(NO$_3$)$_3$ were prepared by dissolving the metal nitrate salts in acetonitrile (MeCN) to obtain 0.1 M solutions. The preparation of crystals suitable for single crystal X-ray Diffraction (scXRD) was prepared by mixing 1 equivalent of the metal nitrate salt with 3 equivalents of 2,2':6',2"-terpyridine (Terpy) (Sigma, 98%) in a 0.5 -1.0 mL of acetonitrile (Sigma, 99.9%). For forming PmTerpy crystals, the hydrated Pm nitrate salt (2 Ci $^{147}$Pm$^{3+}$, 2.1 mg, 0.014 mmol) was carefully mixed with Terpy (9.8 mg, 0.042 mmol) in 0.5 mL of MeCN. Crystals suitable for scXRD formed after ~two hours in a glovebox. Similarly, single crystals on CmTerpy were prepared by mixing the hydrated metal nitrate salt (2.0 mg $^{248}$Cm$^{3+}$, 0.008 mmol) with Terpy (7.4 mg, 0.032 mmol) in 0.5 mL of MeCN. Crystals suitable for scXRD formed overnight.

### Single crystal X-ray diffraction

X-ray diffraction measurements were performed on a Bruker D8 Venture diffractometer equipped with an Iμs 3.0 molybdenum X-ray source (λ = 0.71073 Å). APEX4 software was used for data collection and unit cell determination[30]. The crystal structures were determined using SHELXL or SHELXT software within the OLEX2 graphical user interface (GUI) software[31–33]. Hydrogen atoms bound to carbon were geometrically added at calculated positions. For the radioactive crystals, PmTerpy and CmTerpy, crystals were coated in immersion oil (Type NVH) and then subsequently coated in quick-setting epoxy on a Mitegen Cryoloop. A plastic sheath was then epoxied to the Mitegen Cryoloop base for further radiological protection[34]. This sheath prevents collection of structures at cold temperatures due to icing. Therefore, collections were performed at room temperature. Crystallographic data is located in Supplementary Table 2.

### Single crystal spectroscopy

Single crystals of MTerpy were placed on a microscope slide and contained in Type NVH immersion oil. Absorbance and emission spectra were collected on a CRAIC Technologies 2030PV Pro™ Microspectrophotometer. Emission spectra were collected by exciting with a single wavelength (365 nm or 420 nm).

### Computations

Density Functional Theory (DFT) calculations were performed to further characterize the complexes synthesized. Fully optimized structures were obtained using the meta-Generalized Gradient Approximation (meta-GGA) functional TPSS including Grimme's D3 dispersion corrections along with the Slater-type polarized triple-ζ (STO-TZP). Scalar relativistic effects were included using the ZORA Hamiltonian and implicit solvation effects via the COSMO approximation simulating the complex in acetonitrile. The crystal structure was used as a starting point for the geometry optimization step. All structures were confirmed to be a local minimum via frequency calculations. Default ADF convergence parameters for SCF and geometries were used. The bonding situation was then interrogated through single-point calculations on the optimized geometries where Quantum Theory of Atom in Molecules (QTAIM)

metrics and energy decomposition analyses (EDA) were performed. These single-point calculations were carried out using the hybrid-GGA functional PBE0 to reduce the over-delocalization of the electron density on the complexes. The interaction energies of the complexes were calculated the considering metal nitrate fragments interacting with one and two Terpy ligands for the 1:1 and 2:1 complexes, respectively.

To aid the interpretation of the promethium absorption and emission spectra, the ligand-field DFT (LFDFT) approximation[35] was considered using the same level of theory as in the single-point calculations. Within LFDFT, the multiplet structure for an $f^n$ electron configuration is calculated using the Kohn-Sham orbitals obtained from an average of configuration (AOC) calculation. This allows for cost-efficient multiconfigurational calculations recovering static and varying degrees of dynamic correlation depending on the DFT functional of choice. All calculations were performed in the ADF module within AMS2024[36,37].

## Data availability

CCDC depositions 2366323, 2366324, 2366325, 2366326, 2366327, 2366328, and 2366329 contain the supplementary crystallographic data for this paper. Other crystallographic structures studied in this study are CCDC depositions 142562, 142563, 142564, 142565, and 142566. These data can be obtained free of charge via www.ccdc.cam. ac.uk/data_request/cif, or by emailing data_request@ccdc.cam.ac.uk, or by contacting The Cambridge Crystallographic Data Centre, 12 Union Road, Cambridge CB2 1EZ, UK; fax: +44 1223 336033. The data generated in this study are provided in the Supplementary/Source Data file. All data are available from the corresponding authors upon request. Source data are provided with this paper.

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

## Acknowledgements

This research is supported by the U. S. Department of Energy Isotope Program, managed by the Office of Isotope R&D and Production (FDW). The authors also acknowledge the support of startup funds and Michigan State University (ANG). The authors would like to acknowledge the staff of the REDC processing facility, the promethium production campaign team, and the radiological protection program at ORNL for their assistance in handling, monitoring, and preparation of samples. The authors would also like to thank Dr. Nikki Thiele for use of laboratory resources in preparing the radioactive samples.

## Author contributions

T.B.V, M.E.S, C.J.P, S.K.C, F.D.W executed the synthesis sample preparations, and characterizations. A.J.M., L.H.D, R.T.M performed the isotope processing and dispensing. C.C.B. performed computations. A.N.G, F.D.W conceptualized the project.

## Competing interests

The authors declare no competing interests.
