## [Transparent Peer Review file · Nature Communications]

Probing the Subtle Differences between Promethium and Curium

Corresponding Author: Dr Frankie White

Version 0:

Reviewer comments:

Reviewer #1

(Remarks to the Author)

The authors have made appropriate changes to the manuscript, and I remain strongly supportive of its publication. I do believe the comments of the reviewers have strengthened the manuscript.

Reviewer #2

(Remarks to the Author)

This revised manuscript from White and colleagues focuses on expanding the chemistry of promethium (Pm), an understudied radioactive member of the lanthanide series, and assessing whether Pm is an appropriate surrogate for curium (Cm), which is important for the development of transplutonium actinide chemistry. X-ray diffraction structures for Pm and Cm are identical (other than the metal identity), and subtle differences are noted in metal-ion absorption and luminescence spectra and computational results. These differences are at some points highlighted as minute and in other instances presented as if they are impactful, with the conclusions going as far as describing the Cm complex as featuring enhanced covalent interactions. This example highlights an overarching issue with this version of the manuscript, which is that results are presented inconsistently. At some points this manuscript is a fundamental study that is focused on probing the bonding and spectroscopy of Pm, while at other points the authors are shedding light on Pm/Cm separations based on subtle differences between properties of Pm and Cm terpy congeners. Additionally, while the luminescence results and discussion section has been improved, there are lingering inconsistencies and inaccuracies in the text that raise fundamental scientific questions about this part of the manuscript. Overall, I commend the authors for the technically challenging achievement that is realization of ^{147}Pm and ^{248}Cm crystal structures, and I want to thank them for the time they have committed to revising the manuscript. Upon completing additional revisions, I think this manuscript could be a worthwhile addition to Nature Communications; however, I think additional assessment of the manuscript will be necessary. Aspects of the paper that warrant revision and/or modification are discussed in more detail in the following points.

1. Discussions of Pm radiolysis have largely been removed from the manuscript; however, a follow-up question related to the origin of this phenomenon is what is the radiochemical purity of the ^{147}Pm starting material? The Pm starting material is listed as having a purity >99% in the experimental section, yet no additional context is provided, and the presence of ^{244}Cm in Pm samples would help to explain the accelerated radiolytic effects that are observed.

2. A related question—do Pm crystals lose their long-range order with time (i.e., can data on these samples be collected after more than a week?) If so, and if the Pm does not contain ^{244}Cm , it could be that the crystal color changes are due to the creation of color centers within the crystalline lattice as a result of Pm beta emissions (see Conway and Gruber, *J. Chem. Phys.* 1960, 32, 1586).

3. The luminescence section has been revised but there are still aspects that need to be addressed, such as why has the absorption maxima of terpyridine been shifted by 80 nm for only the Pm complex? If the Pm materials do contain ^{244}Cm (as described above), this could provide an explanation as this isotope is sufficiently radioactive that it could be generating autoradioluminescence effects, which are known for both Cm(III) and Cf(III) (see *Radiochim. Acta* 1991, 52/53, 35 and *J. Chem. Phys.* 1962, 36, 189 for more details about Cm(III) and Cf(III), respectively). There are still multiple mentions of Pm(III) featuring luminescence features in the visible region as well. The first example is the following sentence on pg. 5, "Like Sm^{3+} , Eu^{3+} , Tb^{3+} , and Dy^{3+} , PmTerpy also has luminescence peaks in the visible spectrum. However, these peaks are not well resolved due to the broad band observed at higher energy wavelengths." The second is the following sentence on pg.8, "Likewise, Pm also has luminescence bands in the visible region of the spectrum." These claims do not match with the work of Carnall (*J. Phys. Chem.* 1964, 68, 2351-2357) or Baer (*J. Chem. Phys.* 1973, 59, 2294-2302), and do not make

sense if Jablonski diagrams for the whole Ln series (such as Figure 3 in Acc. Chem. Res. 2009, 42, 542-552) are correct (which would predict only NIR emission for Pm(III)). Additionally, they do not agree with the calculated energy level diagrams in Figures 3 or S1 where energy gaps between excited and ground states are at most 14000 cm⁻¹ (> 700 nm) apart; thus, I would suggest revising or removing these sentences.

4. Finally, in this version of the manuscript there has been a greater emphasis added to the claim that the results here shed light on Pm/Cm purification strategies. The extensive studies that have been conducted on nitrogen donors for lanthanide/minor actinide separations use related nitrogen based heterocycles that feature N,O coordination modes, in contrast to terpy. Moreover, solid-state comparisons, which are what the authors have here from XRD results, do not provide a complete picture about solution state complexation and speciation, so I encourage the authors in the next version to instead focus on strengthening the discussion related to Pm and Cm fundamental bonding and spectroscopic properties as this would enhance the impact and value of the article.

Reviewer #3

(Remarks to the Author)

I recently reviewed this paper for another Nature family journal. As with that review, I will confine my comments to the computational aspects of the paper. The authors have addressed some of my points – we must agree to disagree on the choice of units – but I was rather surprised that the authors have simply ignored several of my previous comments, not even acknowledging them in their rebuttal.

To clarify, I take the text in quotation marks from my first review. In no case can I find any change to the manuscript, nor any text addressing this choice in the authors' response.

"I have never come across the "concentration of the electron density" as a QTAIM term. The authors mean the electron density at the metal-N bond critical points, as indeed they state in the caption to Table 1. I suggest that the term BCP electron density is used throughout."

"On the subject of G and V....I recommend that the authors use G and V to evaluate $-(G/V)$, which is often used as a covalency metric. See Dalton Trans., 2019, 48, 2939–2947 for details."

"As the authors note, the Pauli and electrostatic terms often largely cancel, and overall energy differences are determined by the orbital interaction term. Was any attempt made to understand why the latter term is larger for Cm than the lanthanides, by analysing the Kohn-Sham or localised orbitals?"

The extent to which there is any covalent bonding in the target systems is central to the computational aspects of this paper, yet I still find the messaging unclear. As I noted in my first review, the Discussion section states "only minute differences in bonding are shown by the computational calculations within this Terpy system." Yet in the Conclusions section, we read that the "Cm bond distances are slightly shorter due to enhanced covalent bonding with the nitrogen atoms of the Terpy ligand". I don't believe that the one sentence the authors have added to the Conclusions section is enough to unpick this for the reader – either the computed bonding differences are minute or they are significant enough to evidence larger covalency in the Cm system. This may be just a choice of language – the word "minute" is probably not helping here - but should really be clarified further as it is central to the messaging of the paper.

Reviewer #4

(Remarks to the Author)

This paper reports the synthesis of a novel ¹⁴⁷Pm(III) complex that is isostructural with a known compound that is reported with lanthanides (Ce–Er) and Cm(III). The crystal structures are provided, and bonding trends between the lanthanides and actinides compared. This study includes optical absorption and emission data for ¹⁴⁷Pm(III), which is supported by a complementary computational analysis. The bonding properties are also evaluated using QTAIM and EDA.

The study is notable for its focus on the crystal chemistry and spectroscopy of Pm(III). However, in lines 83-84, the authors suggest the Cm(III) analog was already reported. Please clarify. If not, this certainly adds to the novelty of the work. The connection to separations chemistry feels overstated, as the crystallography, spectroscopic, and computational findings do not directly correlate. This aspect should be deemphasized or omitted. Overall, the study is important, of high quality, and should be published.

Specific Comments:

Introduction:

The discussion regarding ORNL's separations process lacks clarity on whether this issue is specific to ORNL or if alternative separations methods exist that can already achieve the necessary separation. Providing broader context on the challenges faced by the isotope community could strengthen this section.

Overstating Motivations:

The authors mention their aim of gaining insights into the difficult ¹⁴⁷Pm/²⁴⁴Cm separation through the preparation of isostructural compounds with 2,2':6',2''-terpyridine (Terpy). However, the study doesn't sufficiently return to this point, and the collected data doesn't significantly contribute to solving this problem.

The authors also mention their goal to leverage "spectroscopic and computational techniques to further examine the

fundamental differences between the two [Pm and Cm] and potentially provide guidance for the 147Pm/244Cm separation.”

Rather than overstating this goal, the authors could simply emphasize the importance of populating data for this underexplored element to benefit the broader scientific community or merely focus on how understanding just how similar these elements behave is vital. The overselling of a separations theme throughout the manuscript distracts from the strong scientific merit of the study.

Bond Distance Comparisons:

The authors report Pm–N bond distances of 2.612(16) Å and 2.576(9) Å, and compare these to Cm–N bond distances of 2.601(16) Å and 2.551(3) Å, stating that Cm–N bonds are shorter, as predicted. However, for crystallographically derived bond distances to be statistically significant, they must differ by at least 3 times the estimated standard deviation, which these do not. The same issue arises in the discussion of metal–oxygen distances, where the differences are even smaller. This should be addressed in the text.

Absorption Spectra Sentence:

The sentence “Although the major peaks of the Pm³⁺ ion are observed in the solid-state absorption spectra, there are some features worth noting” could be clearer. It may benefit from rewording to clarify the messaging.

Consistency in Element Naming:

The authors switch between using full element names and atomic symbols. This should be consistent throughout the manuscript.

Version 1:

Reviewer comments:

Reviewer #2

(Remarks to the Author)

This newly revised manuscript from White and colleagues focuses on expanding the chemistry of promethium (Pm) and assessing whether Pm is an appropriate surrogate for curium (Cm). I want to start by thanking the authors for the time they have committed to further revising the manuscript, and I think the shift in focus of the manuscript towards a fundamental study that is providing one of the first direct comparisons between Pm and Cm is a welcome development. One of the focuses of updated draft and response to reviewers letter is providing additional clarity related to Pm radiolysis effects, which is critically important as there are not many other research groups who have the capabilities or isotope access to work with 147Pm. As a result, a manuscript such as this one will be a seminal reference for future researchers and currently there are scientific limitations that relate to the discussion of 147Pm and its radioactivity. Thus, my assessment is that the manuscript warrants additional revisions before it is suitable for publication in Nature Communications. Aspects of the paper that warrant revision and/or modification are discussed in more detail in the following points.

1. In my previous review, I asked about the radiochemical purity of the starting material and the authors have indicated that the Pm starting material does not contain 244Cm, which is valuable information to include in the experimental section of the manuscript. Instead, the authors have indicated that the impurities in the 147Pm are from the in-growth of its daughter, 147Sm. However, the presence of 147Sm ($t_{1/2} = 1.068 \times 10^{11}$ yrs.) would not explain the accelerated radiolytic effects that the authors describe on numerous occasions throughout the manuscript. These include the following examples:

- On pg. 3 “However, these crystals quickly turned brown after a short period (~24 hours) likely from radiolytic damage.”
- On pg. 4 “The broadening of the charge transfer band in PmTerpy is likely caused by the radiolytic damage observed in the crystal picture.”
- On pg. 5 “However, while the color of the solution and crystals were initially prickly pear, the color of the PmTerpy crystals quickly turned brown (Figure 2), likely from radiation damage in the organic containing compound. The expected Terpy based transition at 360 nm is significantly broadened and red-shifted nm which could also be attributed to the radioactive nature of Pm.”

The radiolytic effects attributed to Pm warrant additional comment as they occur at a faster rate than those observed with 249Bk in a comparable system (ref. 26), despite 147Pm having a longer half-life, no gamma emissions, and a functionally stable daughter. Moreover, if you consider the beta emission energy of 147Pm (224.1 keV max emission energy per the NNDC) you can calculate the recoil energy for beta emissions from 147Pm atoms, which is 0.84 eV. This is not enough energy to break any of the bonds within a terpyridine heterocycle (for comparison a C-C single bond has a bond energy of approximately 3.6 eV) and the indirect radiation effects that are generated from beta emissions would be unlikely to cause the significant radiolytic effects that have been observed in 24 hours. The Pm starting material is still listed as having a purity >99% in the experimental section and it would be worthwhile to provide information about how this was confirmed. Is this radiochemical purity or is this chemical purity? The presence of other Pm isotopes in small percentages, such as 148Pm ($t_{1/2} = 5.368$ days; high-intensity gamma emissions at 550, 915, and 1465 keV) or 148mPm ($t_{1/2} = 41.3$ days; high-intensity gamma emissions at 288, 414, 550, 599, 630, 725, 915, and 1013 keV), or isobars of neighboring lanthanides, such as 147Nd ($t_{1/2} = 11.03$ days; high-intensity gamma emissions at 91 and 531 keV), would likely lead to the accelerated radiation effects that the authors have noted in structural and spectroscopic results. These isotopic impurities have been found in 147Pm made at HFIR, starting from 146Nd (see the following thesis from UT-Knoxville https://trace.tennessee.edu/utk_gradthes/717/), and it seems probable and worthy of additional comment about how the starting material here compares.

2. A follow-up related to the luminescence results -- in my previous review I noted that Pm visible luminescence claims do not agree with the calculated energy level diagrams in Figures 3 or S1 where energy gaps between excited and ground states are at most 14000 cm⁻¹ (> 700 nm) apart. The authors have subsequently highlighted how in instances where the change in J = 2 that calculations indicate visible emissions are likely to occur from higher energy excited states. Based on Figures 3 or S1, my assessment is that any transition that meet this criterion still has an energy gap that is ≤ 14000 cm⁻¹ and thus would result in NIR emissions. I appreciate the softening of Pm visible luminescence claims in the manuscript and am satisfied with these updates. However, if Figures 3 and S1 are supposed to support claims related to Pm visible luminescence, they should be updated accordingly as well.

3. A smaller suggestion – the updated abstract uses the phrase ‘crystal system’ three times in three consecutive sentences (out of five total in this section). I would suggest an additional round of editing of the abstract to limit the repeated use of this phrase.

Reviewer #4

(Remarks to the Author)

The authors have carefully and thoughtfully considered all comments made by the reviewers. The authors have responded to these comments, justifying inactions and clearly outlining actions taken to address concerns. This manuscript is really great, and I believe the changes made have significantly improved the presentation.

Please consider this work for immediate publication. It is of high quality and will resonate well with the scientific community.

Version 2:

Reviewer comments:

Reviewer #2

(Remarks to the Author)

This updated manuscript from White and colleagues satisfactorily addresses my concerns from the previous version (and I would be excited to read a follow-up study on Pm radiolysis effects within materials). I'm very appreciative of the time and effort the authors put into revisions. This manuscript has been substantively improved via the peer review process, which is due to the authors meaningful engagement with each of the reviewer comments throughout each round of revisions. In my assessment, this manuscript is now suitable for publication in Nature Communications and does not require further revisions.

Response to Reviewers

We are grateful for the reviewers' comments to improve the quality of the manuscript. Below is a response to each of the four reviewers.

Reviewer #1 (Remarks to the Author):

Comment: The authors have made appropriate changes to the manuscript, and I remain strongly supportive of its publication. I do believe the comments of the reviewers have strengthened the manuscript.

Response: We thank the reviewer for their input in improving this manuscript.

Reviewer #2 (Remarks to the Author):

This revised manuscript from White and colleagues focuses on expanding the chemistry of promethium (Pm), an understudied radioactive member of the lanthanide series, and assessing whether Pm is an appropriate surrogate for curium (Cm), which is important for the development of transplutonium actinide chemistry. X-ray diffraction structures for Pm and Cm are identical (other than the metal identity), and subtle differences are noted in metal-ion absorption and luminescence spectra and computational results. These differences are at some points highlighted as minute and in other instances presented as if they are impactful, with the conclusions going as far as describing the Cm complex as featuring enhanced covalent interactions. This example highlights an overarching issue with this version of the manuscript, which is that results are presented inconsistently. At some points this manuscript is a fundamental study that is focused on probing the bonding and spectroscopy of Pm, while at other points the authors are shedding light on Pm/Cm separations based on subtle differences between properties of Pm and Cm terpy congeners. Additionally, while the luminescence results and discussion section has been improved, there are lingering inconsistencies and inaccuracies in the text that raise fundamental scientific questions about this part of the manuscript. Overall, I commend the authors for the technically challenging achievement that is realization of ^{147}Pm and ^{248}Cm crystal structures, and I want to thank them for the time they have committed to revising the manuscript. Upon completing additional revisions, I think this manuscript could be a worthwhile addition to Nature Communications; however, I think additional assessment of the manuscript will be necessary. Aspects of the paper that warrant revision and/or modification are discussed in more detail in the following points.

Comment: Discussions of Pm radiolysis have largely been removed from the manuscript; however, a follow-up question related to the origin of this phenomenon is what is the radiochemical purity of the ^{147}Pm starting material? The Pm starting material is listed as having a purity $>99\%$ in the experimental section, yet no additional context is provided, and the presence of ^{244}Cm in Pm samples would help to explain the accelerated radiolytic effects that are observed.

Response: The radiochemical purity of the material is now elaborated on. The impurities in the Pm-147 are from the ingrowth of its daughter, Sm-147. The Cm-244 in the promethium is below detectable limits to the point where it can be considered negligible. The promethium cannot be provided for research when it still contains Cm-244 because of the dose and its accountability. We reworded areas to further focus on the fundamental chemistry of these elements but want to also stress that this work is indeed to support a very difficult separation. Currently, ORNL is the only producer of Pm-147, in appreciable quantities, but to get it delivered to the public it requires significant processing to remove the Cm-244

after all other fission products have been removed which includes other lanthanides like Ce and Eu, further showing the difficulty of separating the Cm-244 from the Pm-147.

Comment: 2. A related question—do Pm crystals lose their long-range order with time (i.e., can data on these samples be collected after more than a week?) If so, and if the Pm does not contain ^{244}Cm , it could be that the crystal color changes are due to the creation of color centers within the crystalline lattice as a result of Pm beta emissions (see Conway and Gruber, *J Chem. Phys.* 1960, 32, 1586).

Response: Unfortunately, we have not performed a longer duration study to examine the long-range order with time. Currently, synthesis and characterization of our radiological samples takes place in two separate facilities at ORNL, and to reduce material limits of the characterization facilities, it is often required to get the samples back sooner than later. But we will plan to do so in the future. Additionally, considering that the Pm we used doesn't contain Cm-244, this is a possibility.

Comment: 3. The luminescence section has been revised but there are still aspects that needs to be addressed, such as why has the absorption maxima of terpyridine been shifted by 80 nm for only the Pm complex? If the Pm materials do contain ^{244}Cm (as described above), this could provide an explanation as this isotope is sufficiently radioactive that it could be generating autoradioluminescence effects, which are known for both Cm(III) and Cf(III) (see *Radiochim. Acta* 1991, 52/53, 35 and *J. Chem. Phys.* 1962, 36, 189 for more details about Cm(III) and Cf(III), respectively). There are still multiple mentions of Pm(III) featuring luminescence features in the visible region as well. The first example is the following sentence on pg. 5, “Like Sm^{3+} , Eu^{3+} , Tb^{3+} , and Dy^{3+} , PmTerpy also has luminescence peaks in the visible spectrum. However, these peaks are not well resolved due to the broad band observed at higher energy wavelengths.” The second is the following sentence on pg.8, “Likewise, Pm also has luminescence bands in the visible region of the spectrum.” These claims do not match with the work of Carnall (*J. Phys. Chem.* 1964, 68, 2351-2357) or Baer (*J. Chem. Phys.* 1973, 59, 2294-2302), and do not make sense if Jablonski diagrams for the whole Ln series (such as Figure 3 in *Acc. Chem. Res.* 2009, 42, 542-552) are correct (which would predict only NIR emission for Pm(III)). Additionally, they do not agree with the calculated energy level diagrams in Figures 3 or S1 where energy gaps between excited and ground states are at most 14000 cm^{-1} ($> 700\text{ nm}$) apart; thus, I would suggest revising or removing these sentences.

4. Finally, in this version of the manuscript there has been a greater emphasis added to the claim that the results here shed light on Pm/Cm purification strategies. The extensive studies that have been conducted on nitrogen donors for lanthanide/minor actinide separations use related nitrogen based heterocycles that feature N,O coordination modes, in contrast to terpy. Moreover, solid-state comparisons, which are what the authors have here from XRD results, do not provide a complete picture about solution state complexation and speciation, so I encourage the authors in the next version to instead focus on strengthening the discussion related to Pm and Cm fundamental bonding and spectroscopic properties as this would enhance the impact and value of the article.

Response: From the absorption spectrum shown in Figure 2, it is not possible to consider the maximum in the Pm spectrum as the actual peak of the band. This is a saturation effect of the detector. Thus, it cannot be compared to the terpyridine peak shown in the other lanthanide systems. In the first round of responses, we already argued that the literature suggested supports emission in the visible region (and NIR) including the work of Conway. *J. Chem. Phys.* 1960, 32, 1586 at 461, 498, 541, 590, 660, 742 and 830 nm. However, because of the broad band and type of instrument we currently have access to, we have now reworded the statement on promethium luminescence as “Though some reports mention Pm luminescence peaks in the visible region (ref Conway. *J. Chem. Phys.* 1964, 32, 1586), we could not confirm these peaks due to the broad band feature exhibited at higher energies and poor peak resolution.

Other works project Pm does not have peaks in the visible region (ref Acc. Chem. Res. 2009, 42, 542-552). Unfortunately, this area of Pm luminescence remains ambiguous and necessitates further study.”

With respect to the Jablonski diagrams, our diagram does show the possibility for transitions in the visible range. Adding more lines to reflect this would clutter the figure. Regardless, the calculated levels shown in Figure 3 support emission occurring from other 5F_J (and higher-lying) states specially for those transitions involving $|\Delta J| = 2$. As for the diagram cited by the reviewer in Acc. Chem. Res. 2009, 42, 542-552 (which the reviewer does state “if” are correct) does not show the basis for their calculations which makes hard to judge its reliability. However, it is clear that the Pm^{3+} Jablonski diagram presented is incomplete as key low-lying J states are missing and cannot be used for quantitative analysis. Additionally, we are also not trying to present this work as a complete, comprehensive guide on Pm spectroscopy, something that we will be working on in the future. We currently do not have the radiological spectroscopic equipment to flush out all details on the spectroscopy of Pm.

Reviewer #3 (Remarks to the Author):

Comment: I recently reviewed this paper for another Nature family journal. As with that review, I will confine my comments to the computational aspects of the paper. The authors have addressed some of my points – we must agree to disagree on the choice of units – but I was rather surprised that the authors have simply ignored several of my previous comments, not even acknowledging them in their rebuttal.

Response: We apologize if it feels like your comments were not acknowledged. We did take in consideration and responded to many of your comments such as use of the “less” instead of “least” when only comparing two things. We also elaborated on the curiosity of the description of the table in the SI that you pointed out. However, we did accidentally forget to remove the term “concentration” in the manuscript, and that is our fault. However, we did respond to each of your individual comments in the initial Response to Reviewers as we fully appreciate the time taken to review the manuscript. The only comment we acknowledged, but did not change were the units as we believe the current units are more relatable to the general inorganic/f-element community.

To clarify. I take the text in quotation marks from my first review. In no case can I find any change to the manuscript, nor any text addressing this choice in the authors’ response.

Comment: “I have never come across the “concentration of the electron density” as a QTAIM term. The authors mean the electron density at the metal-N bond critical points, as indeed they state in the caption to Table 1. I suggest that the term BCP electron density is used throughout.”

Response: We understand the reviewer’s concern and modified accordingly. Though, we would like to clarify that the electron density can be considered as accumulated in a defined point in the molecular space from a topological standpoint. However, we realize that its use among the f-element community is rather uncommon.

Comment: “On the subject of G and V...I recommend that the authors use G and V to evaluate $-(G/V)$, which is often used as a covalency metric. See Dalton Trans., 2019, 48, 2939–2947 for details.”

Response: We appreciate the suggestion, but we think the insight gained by $-(G/V)$ is marginal as the systems studied are very similar in bonding. We have previously used the $|V|/G$ metric as it ranges

between 1 and 2 for partial covalent bonds. In our systems, this metric shows 88%-91% polarizability of the bond towards the ligand, which does not provide much information other than what it is already known from Table 1.

Comment: “As the authors note, the Pauli and electrostatic terms often largely cancel, and overall energy differences are determined by the orbital interaction term. Was any attempt made to understand why the latter term is larger for Cm than the lanthanides, by analysing the Kohn-Sham or localised orbitals?”

Response: Yes. The results were further analyzed through ETS-NOCV but differences of less than 3-4 kcal/mol (which are marginal) are difficult to see and were not included in the manuscript. It is rather unusual to see such small differences between a lanthanide and an actinide, but not uncommon for a half-filled systems as they tend to be less prone to charge transfer interactions.

Comment: The extent to which there is any covalent bonding in the target systems is central to the computational aspects of this paper, yet I still find the messaging unclear. As I noted in my first review, the Discussion section states “only minute differences in bonding are shown by the computational calculations within this Terpy system.” Yet in the Conclusions section, we read that the “Cm bond distances are slightly shorter due to enhanced covalent bonding with the nitrogen atoms of the Terpy ligand”. I don’t believe that the one sentence the authors have added to the Conclusions section is enough to unpick this for the reader – either the computed bonding differences are minute or they are significant enough to evidence larger covalency in the Cm system. This may be just a choice of language – the word “minute” is probably not helping here - but should really be clarified further as it is central to the messaging of the paper.

Response: We agree with the reviewer. In the discussion section we have clarified that the minute difference refers to having orbital interaction differences of less than 3 kcal/mol between Cm and Pm. Similarly, the conclusions have been also modified where an attribution to enhanced covalency has been removed to avoid confusion.

Reviewer #4 (Remarks to the Author):

This paper reports the synthesis of a novel $^{147}\text{Pm(III)}$ complex that is isostructural with a known compound that is reported with lanthanides (Ce–Er) and Cm(III). The crystal structures are provided, and bonding trends between the lanthanides and actinides compared. This study includes optical absorption and emission data for $^{147}\text{Pm(III)}$, which is supported by a complementary computational analysis. The bonding properties are also evaluated using QTAIM and EDA.

Comment: The study is notable for its focus on the crystal chemistry and spectroscopy of Pm(III). However, in lines 83-84, the authors suggest the Cm(III) analog was already reported. Please clarify. If not, this certainly adds to the novelty of the work. The connection to separations chemistry feels overstated, as the crystallography, spectroscopic, and computational findings do not directly correlate. This aspect should be deemphasized or omitted. Overall, the study is important, of high quality, and should be published.

Response: We clarify that both the Pm and Cm structures were unknown by adding this this statement in the Synthesis and Crystallography section: “The isostructural ^{147}Pm and Cm analogs along with several other unreported lanthanide analogs...”.

Comment: Specific Comments:

Introduction:

The discussion regarding ORNL's separations process lacks clarity on whether this issue is specific to ORNL or if alternative separations methods exist that can already achieve the necessary separation. Providing broader context on the challenges faced by the isotope community could strengthen this section.

Response: We now mention that ORNL is currently the only producer of ^{147}Pm . Furthermore, we have shifted the manuscript to a more fundamental focus to examine why the separation is difficult versus how we are trying to improve the separation.

Comment: Overstating Motivations:

The authors mention their aim of gaining insights into the difficult $^{147}\text{Pm}/^{244}\text{Cm}$ separation through the preparation of isostructural compounds with 2,2':6',2''-terpyridine (Terpy). However, the study doesn't sufficiently return to this point, and the collected data doesn't significantly contribute to solving this problem.

The authors also mention their goal to leverage "spectroscopic and computational techniques to further examine the fundamental differences between the two [Pm and Cm] and potentially provide guidance for the $^{147}\text{Pm}/^{244}\text{Cm}$ separation."

Rather than overstating this goal, the authors could simply emphasize the importance of populating data for this underexplored element to benefit the broader scientific community or merely focus on how understanding just how similar these elements behave is vital. The overselling of a separations theme throughout the manuscript distracts from the strong scientific merit of the study.

Response: We have taken this into consideration. Rather than probing the similarities of Pm and Cm from an improving separations viewpoint, we have changed the wording and focus to examining the fundamental properties to understand the similarities between the two elements due to their difficult separation from each other. Verbiage in the abstract and throughout the introduction now reflects that the difficulty of separating Pm and Cm serves as the inspiration for examining the fundamental properties between the two elements. We also mention that the spectroscopy could be used to aid in a separation and added a work that was recently published to show that Cm luminescence can be utilized on a column.

Comment: Bond Distance Comparisons:

The authors report Pm–N bond distances of 2.612(16) Å and 2.576(9) Å, and compare these to Cm–N bond distances of 2.601(16) Å and 2.551(3) Å, stating that Cm–N bonds are shorter, as predicted. However, for crystallographically derived bond distances to be statistically significant, they must differ by at least 3 times the estimated standard deviation, which these do not. The same issue arises in the discussion of metal–oxygen distances, where the differences are even smaller. This should be addressed in the text.

Response: The last paragraph of the Synthesis and Crystallography section now has a sentence that reads "Albeit not statistically significant given the bond errors, this hypothesis appears supported in the MTerpy system..." Additionally in the Discussion, we state that "While it was expected for Cm to have somewhat shorter bonds to the nitrogen containing Terpy ligand, the differences are not significant."

Comment: Absorption Spectra Sentence:

The sentence “Although the major peaks of the Pm³⁺ ion are observed in the solid-state absorption spectra, there are some features worth noting” could be clearer. It may benefit from rewording to clarify the messaging.

Response: The sentence has been clarified to read “Although the solid-state absorption spectra of the Pm³⁺ ion was as expected, there are aspects worth discussing.”

Comment: Consistency in Element Naming:

The authors switch between using full element names and atomic symbols. This should be consistent throughout the manuscript.

Response: We use the full names in the abstract and then refer to the atomic symbols throughout the rest of the manuscript. We use “Pm” and “Cm” in circumstances regarding all isotopes of promethium or curium and “¹⁴⁷Pm” and “²⁴⁴Cm” where regarding those specific isotopes.

Reviewer #2 Comments

This newly revised manuscript from White and colleagues focuses on expanding the chemistry of promethium (Pm) and assessing whether Pm is an appropriate surrogate for curium (Cm). I want to start by thanking the authors for the time they have committed to further revising the manuscript, and I think the shift in focus of the manuscript towards a fundamental study that is providing one of the first direct comparisons between Pm and Cm is a welcome development. One of the focuses of updated draft and response to reviewers letter is providing additional clarity related to Pm radiolysis effects, which is critically important as there are not many other research groups who have the capabilities or isotope access to work with ^{147}Pm . As a result, a manuscript such as this one will be a seminal reference for future researchers and currently there are scientific limitations that relate to the discussion of ^{147}Pm and its radioactivity. Thus, my assessment is that the manuscript warrants additional revisions before it is suitable for publication in Nature Communications. Aspects of the paper that warrant revision and/or modification are discussed in more detail in the following points.

Comment 1: In my previous review, I asked about the radiochemical purity of the starting material and the authors have indicated that the Pm starting material does not contain ^{244}Cm , which is valuable information to include in the experimental section of the manuscript. Instead, the authors have indicated that the impurities in the ^{147}Pm are from the in-growth of its daughter, ^{147}Sm . However, the presence of ^{147}Sm ($t_{1/2} = 1.068 \times 10^{11}$ yrs.) would not explain the accelerated radiolytic effects that the authors describe on numerous occasions throughout the manuscript. These include the follow examples:

- On pg. 3 “However, these crystals quickly turned brown after a short period (~24 hours) likely from radiolytic damage.”
- On pg. 4 “The broadening of the charge transfer band in PmTerpy is likely caused by the radiolytic damage observed in the crystal picture.”
- On pg. 5 “However, while the color of the solution and crystals were initially prickly pear, the color of the PmTerpy crystals quickly turned brown (Figure 2), likely from radiation damage in the organic containing compound. The expected Terpy based transition at 360 nm is significantly broadened and red-shifted nm which could also be attributed to the radioactive nature of Pm.”

The radiolytic effects attributed to Pm warrant additional comment as they occur at a faster rate than those observed with ^{249}Bk in a comparable system (ref. 26), despite ^{147}Pm having a longer half-life, no gamma emissions, and a functionally stable daughter. Moreover, if you consider the beta emission energy of ^{147}Pm (224.1 keV max emission energy per the NNDC) you can calculate the recoil energy for beta emissions from ^{147}Pm atoms, which is 0.84 eV. This is not enough energy to break any of the bonds within a terpyridine heterocycle (for comparison a C-C single bond has a bond energy of approximately 3.6 eV) and the indirect radiation effects that are generated from beta emissions would be unlikely to cause the significant radiolytic effects that have been observed in 24 hours. The Pm starting material is still listed as having a purity >99% in the experimental section and it would be worthwhile to provide information about how this was confirmed. Is this radiochemical purity or is this chemical purity? The presence of other Pm isotopes in small percentages, such as ^{148}Pm ($t_{1/2} = 5.368$ days; high-intensity gamma emissions at 550, 915, and 1465 keV) or ^{148m}Pm ($t_{1/2} = 41.3$ days; high-intensity gamma emissions at 288, 414, 550, 599, 630, 725, 915, and 1013 keV), or isobars of neighboring lanthanides, such as ^{147}Nd ($t_{1/2} = 11.03$ days; high-intensity gamma emissions at 91 and 531 keV), would likely lead to the accelerated radiation effects that the authors have noted in structural and spectroscopic results. These isotopic impurities have been found in ^{147}Pm made at HFIR, starting from ^{146}Nd (see the following thesis from UT-Knoxville https://trace.tennessee.edu/utk_gradthes/717/), and it seems probable and worthy of additional comment

about how the starting material here compares.

Response: I think deep diving into the radiochemical processes that are going on in the Pm crystal structure is starting to go beyond the scope of this manuscript. An entire paper could be dedicated to the examining the radioactive effects of Pm in molecular complexes. However, although not as high as Bk-249 (specific activity ~1649 Ci/g), the specific activity of Pm-147 (specific activity ~ 928 Ci/g) is still relatively high when compared to other commonly used research isotopes in f-element chemistry (i.e. Am-243, Cf-249, Pu-239 though these are alpha emitters) and other beta emitters (i.e. Ni-63 specific activity ~15 Ci/g, Sr-90 specific activity ~ 138 Ci/g). It is noted that Pm is intensely radioactive in other sources to the point where it can self-darken inorganic compounds as in stated in Conway et al. *J. Chem. Phys.* 32, 1586–1587 (1960). In this work, they believe “*The darkening may be bleached with heat or light. We believe the darkening is due to the trapping of free electrons into the crystal defects (color centers). The source of the electrons is the soft β -’s emitted by the decaying Pm¹⁴⁷.*” This could be the reasoning for the darkening; however, this paper is also the work that reports peak groupings of Pm luminescence in the visible region which the reviewer contests. Because we cannot state with certainty the exact causes or method by which the crystals darken, we believe it is best not to over-comment. The stable lanthanides and relatively low activity Cm-248 compounds do not exhibit these traits, and the only difference between these and the Pm-147 structure is the associated radioactivity. Therefore, we believe it is safe to conclude the radioactivity of the Pm-147 has something to do with these effects. Though an understandable comparison to Bk-249 structures in the past, we have commented before that these are different structures and could be affected by radiolysis differently. The comparison to high activity Bk-249 should not minimize the fact that Pm-147 is still highly radioactive itself and has a relatively short half-life. Additionally, while the radioactivity may not break bonds, it could disrupt the ordering in the crystal lattice such as the co-crystallized outer-sphere acetonitrile solvent molecules, but this would also be speculating which we prefer not to do.

Regarding the purity, the Pm-147 is greater than 99% purity both radiochemical and chemical purity as there is less than 1% of Sm-147, other isotopes, and stable elements. Because the Pm-147 used is mined from the Pu-238 waste stream and not from irradiation of Nd, though not detectable, that doesn’t mean that there are absolutely no atoms of other fission products like cerium and europium isotopes or Cm-244 in the Pm-147 we used that could also potentially affect the Pm-147 crystals. Regarding the technique used to determine the purity, all I can comment on is that the technique in which the purity was determined is a particular method of ICP-MS analysis and is done by another organization. However, I do not know this method nor were we involved in this analysis and simply used the Pm-147 as received.

Comment 2: A follow-up related to the luminescence results -- in my previous review I noted that Pm visible luminescence claims do not agree with the calculated energy level diagrams in Figures 3 or S1 where energy gaps between excited and ground states are at most 14000 cm⁻¹ (> 700 nm) apart. The authors have subsequently highlighted how in instances where the change in J = 2 that calculations indicate visible emissions are likely to occur from higher energy excited states. Based on Figures 3 or S1, my assessment is that any transition that meet this criterion still has an energy gap that is \leq 14000 cm⁻¹ and thus would result in NIR emissions. I appreciate the softening of Pm visible luminescence claims in the manuscript and am satisfied with these updates. However, if Figures 3 and S1 are supposed to support claims related to Pm visible luminescence, they should be updated accordingly as well.

Response: We have updated the figures in the manuscript and SI to reflect these transitions.

Comment 3: A smaller suggestion – the updated abstract uses the phrase ‘crystal system’ three times in three consecutive sentences (out of five total in this section). I would suggest an additional round of editing of the abstract to limit the repeated use of this phrase.

Response: The ending of the abstract now reads as “Analysis of the isostructural compounds via single

crystal X-ray diffraction and quantum theory of atoms in molecules (QTAIM) revealed that bonding between Pm and Cm is quite similar in this particular structure type. The small differences observed in the analysis of these two elements in this isostructural series shed light on the difficulty required to separate the elements from each other. More so, this study develops the fundamental chemistries of two rare elements in the solid state and experimentally portrays the often-omitted position of Pm within the lanthanide series.”

Reviewer #4

Comment: The authors have carefully and thoughtfully considered all comments made by the reviewers. The authors have responded to these comments, justifying inactions and clearly outlining actions taken to address concerns. This manuscript is really great, and I believe the changes made have significantly improved the presentation.

Please consider this work for immediate publication. It is of high quality and will resonate well with the scientific community.

Response: We thank the reviewer for their feedback in improving this manuscript.